# C-Reactive Protein and Neutrophil/Lymphocytes Ratio: Prognostic Indicator for Doubling Overall Survival Prediction in Pancreatic Cancer Patients

**DOI:** 10.3390/jcm8111791

**Published:** 2019-10-25

**Authors:** Konstantin Schlick, Teresa Magnes, Florian Huemer, Lukas Ratzinger, Lukas Weiss, Martin Pichler, Thomas Melchardt, Richard Greil, Alexander Egle

**Affiliations:** 13rd Medical Department with Hematology and Medical Oncology, Hemostaseology, Rheumatology and Infectious Diseases, Paracelsus Medical University, 5071 Salzburg, Austria; K.Schlick@salk.at (K.S.); t.magnes@salk.at (T.M.); f.Huemer@salk.at (F.H.); l.ratzinger@salk.at (L.R.); lu.weiss@salk.at (L.W.); t.melchardt@salk.at (T.M.); A.egle@salk.at (A.E.); 2Salzburg Cancer Research Institute (SCRI) Laboratory for Immunological and Molecular Cancer Research (LIMCR) Center for Clinical Cancer and Immunology Trials (CCCIT), 5071 Salzburg, Austria; 3Division of Clinical Oncology, Department of Medicine, Medical University of Graz, 8036 Graz, Austria; martin.pichler@medunigraz.at; 4Department of Experimental Therapeutics, The University of Texas MD Anderson Cancer, Houston, TX 77030, USA

**Keywords:** pancreatic cancer, prognostais, CRP, Neutrophil/Lymphocytes ratio, OS

## Abstract

Background: Despite modern chemotherapy regimens, survival of patients with locally advanced/metastatic pancreatic cancer remains dismal. Long-term survivors are rare and there are no prognostic scores to identify patients benefitting most from chemotherapy. Methods: This retrospective study includes 240 patients with pancreatic cancer who were treated in a primary palliative setting between the years 2007 to 2016 in a single academic institution. Survival rates were analyzed using the Kaplan–Meier method. Prognostic models including laboratory and clinical parameters were calculated using Cox proportional models in univariate and multivariate analyses. Results: Median age at diagnosis was 67 years (range 29–90 years), 52% were female and a majority had an ECOG performance status of 0 or 1. Locally advanced pancreatic cancer was diagnosed in 23.3% (*n* = 56) and primary metastatic disease in 76.7% (*n* = 184) of all patients. Median overall survival of the whole study cohort was 8.3 months. Investigating potential risk factors like patient characteristics, tumor marker or inflammatory markers, multivariate survival analysis found CRP (c-reactive protein) and NLR (neutrophil to lymphocyte ratio) elevation before the start of palliative chemotherapy to be independent negative prognostic factors for OS (overall survival) (*p* < 0.001 and *p* < 0.01). Grouping patients with no risk factor versus patients with one or two of the above mentioned two risk factors, we found a median OS of 16.8 months and 9.4 months (*p* < 0.001) respectively. By combining these two factors, we were also able to identify pancreatic cancer patients that were more likely to receive any post first line therapy. These two risk factors are predictive for improved survival independent of disease stage (III or IV) and applied chemotherapy agents in first line. Conclusion: By combining these two factors, CRP and NLR, to create a score for OS, we propose a simple, new prognostic tool for OS prediction in pancreatic cancer.

## 1. Introduction

Pancreatic cancer (PC) is currently associated with one of the lowest survival rates among malignancies, resulting in an overall mortality after 5 years of follow up of 95% [1]. Unfortunately, prognosis is dismal even though novel therapy strategies were implemented in clinical routine [2]. PC is the fourth leading cause of cancer-related death in the US and the only potential chance for cure remains radical resection [3]. However, only 15% to 20% of all newly diagnosed patients can be considered for surgical resection, resulting in an improved survival of up to 24 months [4]. Inoperable, advanced, and metastatic disease shows significantly shorter survival. Current options to predict the duration of OS (overall survival) in PC remain unsatisfying. Only a small set of parameters like histopathology, imaging of tumor load, and tumor marker are frequently used in clinical practice for decision-making. Risk scores do not uniformly define therapy responders or long term survivors [5,6,7,8].

Inflammation is thought to influence carcinogenesis through DNA damage and activation of intracellular signaling pathways. Inflammatory biomarkers like lymphocytes play a role in lymphocyte-mediated anti-tumor activity by inducing cell apoptosis and inhibiting cancer cell proliferation and migration in pancreatic cancer [9]. Therefore a high neutrophil to lymphocyte ratio (NLR) resulting either from neutrophilia or relative lymphopenia or both seems also to be associated with survival of pancreatic cancer patients [10]. Many studies have focused on NLR as a negative predictor on OS, not only in PC, but in various cancers such as lung cancer, esophageal, gastric, and colorectal cancer [11]. Notably, Bailey et al conducted a clustering of pancreatic cancer RNA sequencing data and classified an immunogenic subtype of PC showing a significant immune infiltrate, which may be associated with pancreatitis and carcinogenesis [12]. Furthermore, association of chronic inflammation, seen in elevated CRP (c- reactive protein) levels, with shorter OS is described in several entities, like colorectal cancer, urothelial cancer, and lymphomas [13].

The involvement of all these inflammatory biomarkers support the theory that recurrent pancreatitis maybe leading to activation of inflammatory pathways, chronic disease development, and therefore cellular injury [14]. Such an inflammatory status was shown to facilitate oncogenetic KRAS mutations and altered tumor suppressors (p53 and p16), resulting in the development of PC leading to tumor initiation and tumor promotion [14,15,16].

Several studies investigated the relationship of tumor markers (Ca19-9) elevations to survival rates in pancreatic cancer patients, showing a correlation of higher tumor markers with worse OS [17]. Furthermore classical patient characteristics, chronic inflammation markers, and habitual disposition like body mass index were analyzed for possible risk factors, showing a clinical relevant connection between inflammation and adiposity to shorter OS in gastrointestinal cancer patients including pancreatic cancer [18]. 

As there are no standardized and validated prognostic risk scores for pancreatic cancer, the question posed in this study is if a combination of the already published inflammation markers, based on blood parameters, could provide clinicians with a better prognostic tool for stratification of patients’ mortality risks. 

## 2. Materials and Methods

### 2.1. Patient Selection and Data Acquisition

We aimed to investigate risk factors for outcome in pancreatic cancer receiving chemotherapy in a palliative setting (advanced or metastatic disease). From an academic retrospective observational cohort treated at the IIIrd Medical Department of the Paracelsus Medical University Salzburg, 367 patients with diagnosis of PC were screened and 127 patients had to excluded due to receiving either only adjuvant therapy at the time of this analysis, or no therapy at all due to detrimental performance status at pancreatic cancer diagnosis. For proposing a prognostic score in primary palliative setting, we identified 240 patients with locally advanced and metastatic disease for our analyses in order to form a most homogeneous study cohort by disease stage and treatment goal. 

All patients over 18 years of age diagnosed between January 2007 and May 2016 with pathologically- or imaging-confirmed pancreatic adenocarcinoma in locally advanced (Stage III) or metastatic (Stage IV) disease setting were eligible for inclusion. All patients underwent primary palliative chemotherapy according to local standards (see Table 1 for detailed list of chemotherapy regimens used). To identify prognostic factors for patients, we not only focused on chronic inflammation factors, but also on tumor related factors and patients’ individual characteristics. Patients were evaluated for gender, age, complete blood count, and serum markers [19]. The laboratory data included red and white blood count, including neutrophils, lymphocytes and monocytes, platelets, levels of tumor markers (CEA and Ca19-9), and CRP levels. In addition, the following derived variables were generated: neutrophil to lymphocyte, lymphocyte to monocyte, and platelet derived to lymphocyte ratio. All factors were determined prior to start of palliative chemotherapy and performed as a part of routine clinical practice.

Tumor response was defined using CT scans and classified into partial response, stable or progressive disease according to WHO criteria, and by tumor marker response [20]. Progression free survival (PFS) was defined as the time from start of a treatment until the date of progress or death, and overall survival (OS) was defined as time of diagnosis of PC until death from any cause. All clinico-pathological data were retrieved from medical records at the Department of Oncology, as well as from pathology records from the Institute of Pathology at the same institution. Treatment decisions were made according to the current ESMO guidelines at the time of treatment [21] and followed the tumor board decision. This analysis was approved the Ethics Committee of the province of Salzburg, Austria. 

### 2.2. Statistics

Data analysis for this retrospective study was descriptive in nature and presented in medians and ranges with 95% CI. OS estimates were obtained using Kaplan–Meier (KM) method and tested for significance using the log rank test. The median follow-up duration was measured by reverse KM estimator. For association of clinical pathological parameters with OS, we initially performed a univariate assessment of predefined prognostic parameters comparing survival curves by the use of Cox regression univariate analyses, including hazard ratio (HR) with a 95% confidence interval (CI). For metric parameters, the upper limit of normal was taken as the cut-off value; if no upper limit was defined, a best cut-off was calculated based on the receiver operating characteristics (ROC) analyses and the Youden Index J, which represents the maximum sensitivity and specificity for all cut points in the ROC curve [22]. Multivariate Cox regression analysis for OS was then calculated using all significant variables from the univariate analyses. Statistical analyses were performed with a statistical software package (SPSS, version 21 IBM Corp). 

For sub-group analyses, Person´s chi-squared test was used to determine whether there is a significant difference between the expected frequencies and the observed frequencies in one or more categories.

A *p*-value of <0.05 was considered statistically significant. 

In order to avoid multiple testing errors, not only an MVA, but also an additional simple Bonferroni Correction Test was performed for an analysis with 11 parameters, resulting in a corrected minimal significance level of *p* < 0.004.

## 3. Results

### 3.1. Patient Characteristics 

Within our study cohort of 240 pancreatic cancer patients, patients had a median age at diagnosis of 67 years (range 29–90 years). Fifty-two percent were female, and the majority had Eastern Cooperative Oncology Group performance status (ECOG)of 0 or 1. Median overall survival of the whole study cohort was 8.3 months. Locally advanced pancreatic cancer was present in 23.3% (*n* = 56) and primary metastatic disease was observed in 76.7% (*n* = 184) of all patients. There was no statistically significant difference in OS regarding these two groups, 7.4 vs. 8.9 months (*p* = 0.34). All patients treated with at least one dose of chemotherapy were eligible for our analyses. Patient characteristics are shown in detail in Table 1.

### 3.2. Treatment

Palliative first line therapy included FOLFIRINOX in 42.5% of the patients (*n* = 102) and gemcitabine in combination with nab-pacliataxel or oxaliplatin in 12.9% of the patients (*n* = 31) and 17% of the patients (*n* = 41), respectively. A monotherapy with gemcitabine was applied in 21% of the patients (*n* = 51). The rest, 6.2% of the patients, (*n* = 15) received various chemotherapeutic agents (see Appendix A).

More than half (57.5%) of all patients (*n* = 138) received 2nd line chemotherapy and a 3rd line was given in 27.5% of all patients (*n* = 66) (See Appendix A).

Patients receiving only 1st line palliative chemotherapy had statistically significantly higher inflammation markers including elevated CRP (*p* < 0.004), leukocytosis (*p* < 0.001), neutrophilia (*p* < 0.002), and elevated NLR (*p* < 0.011) compared to patients receiving a 2nd and 3rd line therapy according to Pearson´s chi-squared testing. 

### 3.3. Prognostic Factor Analyses

To investigate whether clinico-pathological factors, tumor markers, and chronic inflammation were associated with the clinical outcome of PC patients, and to differentiate between shorter and longer OS univariate, multivariate Cox proportional analyses were performed. Univariate Cox proportional analyses identified leukocytosis (*p* < 0.001), neutrophilia (*p* < 0.001), monocytosis (*p* < 0,001), elevated NLR (*p* < 0.001), elevated LMR (*p* < 0,009), elevated CRP (*p* < 0.001), and elevated tumor marker CEA (*p* < 0.019) as significantly associated with shorter OS (see Table 2 for further details).

To determine the independent prognostic values for OS, multivariate analysis was conducted. In the multivariate analysis that included all six factors significantly associated with survival in univariate analyses, we identified elevated NLR (defined as >upper limit of no normal; ULN) and elevated CRP (defined as >ULN), as remaining independent prognostic factors for shorter OS. In order to confirm our findings from the MVA and to further exclude multiple testing errors, a simple Bonferroni Correction Test was performed, leading to a corrected significance level of *p* < 0.004. Using this level of stringency, we again found elevated CRP and NLR to be statistically significantly associated with OS.

We found a weak (R^2^ = 0.25) but significant association between CRP and NLR elevation (*p* < 0.001). Together with the large spread of individual datapoints in the scattergram (Appendix A) and the results from the multivariate model, this suggests that significant independent information may be derived from both variables. 

By combining the two risk factor groups, we created a prognostic score (see Figure 1) Of the 145 patients with complete data available for all variables, 45 patients had none of the above-mentioned risk factors identified in MVA, 65 patients had one, and 35 had both risk factors. The median OS for all patients was 11.2 months. By patients with any factor versus those with zero risk factors, we found median OS of 9.4 months and 16.8 months respectively. At 12 and 24 months, 51% and 16% of patients in the zero risk factor group were alive compared to 26% and 4% respectively in the one and two risk factor group. Survival differences among groups were statistically significant (HR = 1.9; 95% CI = (1.3–2.7); *p* < 0.001) and demonstrated a likely clinically meaningful survival benefit by a factor of close to two. Further, in Figure 2 we show patients with zero vs. one vs. two risk factors. 

The 145 patients with complete data available for all variables. Forty-five patients had none of the above-mentioned risk factors, 65 patients had one, and 35 had both risk factors. The median OS for all patients was 8.3 months. Patients with zero risk factors had 11.3 months compared to 7.1 months with one and 3.8 months with two risk factors, respectively. Survival differences among groups were statistically significant (HR = 1.558; 95% CI = (1.26–1.95); *p* < 0.001).

## 4. Discussion

Regarding the management of PC with several treatment strategies having, level I A evidence ranging from polychemotherapy (FOLFIRNOX), to chemo-doublets (gemcitabine and nab-paclitaxel), to monotherapy with gemcitabine, there are only few measures with meaningful impact on the prediction of OS. Furthermore, each of these regiments come with a different set of expected toxicities.

Assessment of patient risks using histology of tumor, extent of its spread, and elevated tumor marker are unsatisfactory. Our study is one of the largest single center retrospective analysis of a real world, relatively unselected patient collective with regards to co-morbidities, age, and performance status. There are no oncologic practitioners in private practice in the Austrian health care system. The treatment of patients with pancreatic cancer in Salzburg and the surrounding regions is thus centralized to our department. Given the Austrian health insurance system, this analysis is also likely to be less biased regarding insurance status or other socioeconomic factors than one might assume in some international randomized clinical trials. 

With this analysis, we propose an easily applicable risk score, derived from circulating biomarkers (commonly available laboratory parameters) to help in patient stratification and clinical decision making, as there are no standardized and validated prognostic risk scores for pancreatic cancer patients. 

In this study, we addressed the prognostic significance of CRP and NLR in patients with pancreatic cancer by comparing various factors in uni- and multivariate analyses. 

The ratio of neutrophils and lymphocytes is thought to be a surrogate marker for ongoing inflammation [23]. Systemic inflammation and immune response play a crucial role in cancer cell growth and NLR is possibly a very simply-assessable indicator for prognosis in various solid tumors. A high NLR already has been shown to have a negative influence on overall survival in various cancer entities [24,25,26].

Further markers of chronic inflammation, e.g., elevated CRP levels, were associated with shorter OS as described in several entities, like colorectal cancer, urothelial cancer, and lymphomas [13].

With our retrospective analyses, we show that elevated NLR, as a surrogate marker for subclinical inflammation and CRP elevation, correlate statistically significant with poor prognosis in PC independent of applied chemotherapy regiments. Additional subgroup analyses defined which patients are more likely to receive 2^nd^ and 3^rd^ line chemotherapy. Patients undergoing post-first-line therapy had less elevated inflammation makers at the time point of pancreatic cancer diagnosis. Therefore, assessment of inflammation markers at diagnosis allowed to distinguish a group with a very bad prognosis using chemotherapy. 

With respect to the relationship between NLR and high CRP levels, our multivariate analyses indicated that these two parameters were independent predictive factors for poor survival. In addition, we used a second independent approach to tackle multiple testing biases, with the same result. 

Targeting inflammatory reaction in PDAC by inhibiting or depleting pro-tumor elements and by engaging the potential of inflammatory cells to acquire anti-tumor activity has garnered strong research and clinical interest. Strategies to intervene on inflammation for providing therapeutic benefit are currently undergoing in Phase 1 and 2 trials [27,28].

We present this type of analyses to guide treatment decisions in bad risk groups and suggests other research groups to validate our results. The combination of the two parameters into a prognostic score predicted OS with a HR of close to two, suggesting an option for prediction of a clinically meaningful benefit, with a rough doubling of one and two year survival chances of patients without these inflammatory risk factors.

## 5. Main Message

NLR and high CRP levels proved independent predictors of overall survival in patients receiving palliative chemotherapy for pancreas carcinoma with advanced or metastatic disease.

We thus propose that these easily accessible inflammation markers might be used to identify patients for different types of treatment approaches.

## Figures and Tables

**Figure 1 jcm-08-01791-f001:**
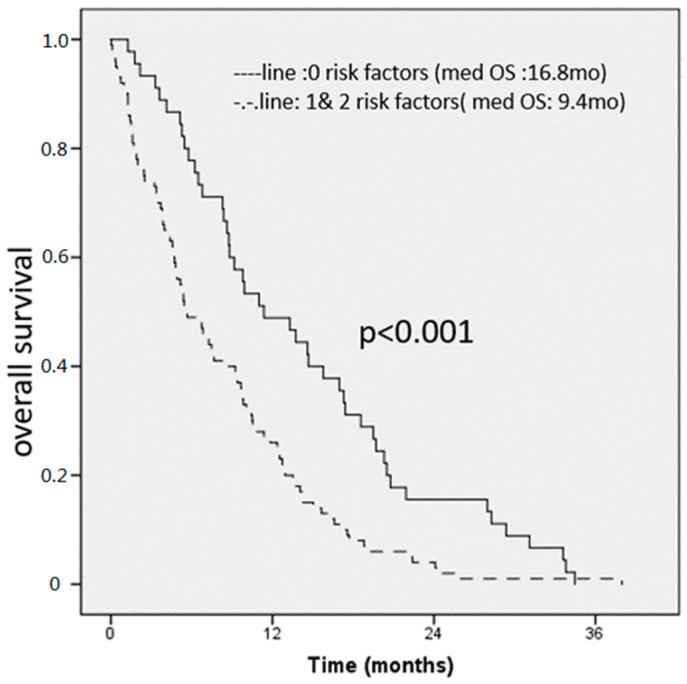
Overall survival score for all patients (n0148). An elevated CRP and NLR were found to be independent prognostic factors for OS, patients with 0 risk factors had 16.8 months, compared to 9.4 months for patients with 1 or 3 risk factors respectively.

**Figure 2 jcm-08-01791-f002:**
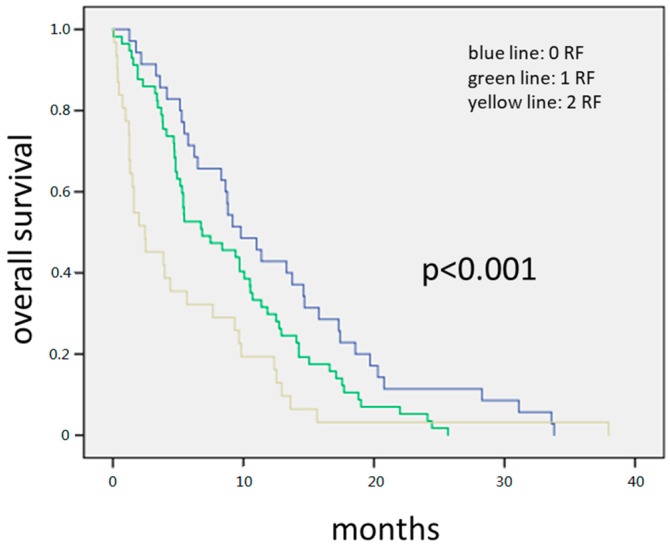
The 145 patients with complete data available for all variables. Forty-five patients had none of the above-mentioned risk factors, 65 patients had one, and 35 had both risk factors. The median OS for all patients was 8.3 months. Patients with zero risk factors had 11.3 months compared to 7.1 months with one and 3.8 months with two risk factors, respectively. Survival differences among groups were statistically significant (HR = 1.558; 95% CI = (1.26–1.95); *p* < 0.001).

**Table 1 jcm-08-01791-t001:** Characteristics of the Salzburg Pancreas cohort 240 patients (2006–2017).

Characteristic	Score (*N* = 240)
Age—no. of years	
Median	67
Range	29–90
Sex—no. (%)	
Male	114 (48%)
Female	126 (52%)
Tumor stage at diagnosis—no. (%)	
Locally advanced	56 (23.3%)
Metastatic	184 (76.7%)
Palliative 1st line protocol—no. (%)	
FOLFIRINOX:	102 (42.5%)
Gemcitabine/Nab-Paclitaxel	31(12.9%)
Gemcitabine	51 (21%)
GEMOX	41 (17%)
Other	15 (6.2%)
Chemotherapy beyond 1st line protocol—no. (%)	
2nd line palliative chemotherapy	138 (57.7%)
3rd line palliative chemotherapy	66 (27.5%)

Legend: FOLFIRINOX (5-Fu, oxaliplatin, irinotecan), GEMOX (gemcitabine, oxaliplatin).

**Table 2 jcm-08-01791-t002:** Prognostic factors for overall survival.

Variable (above ULN)		Univariate			Multivariate		
		HR (95% CI)	*P* (1)	*n*	HR (95% CI)	*P* (1)	*n*
**Gender**	Male vs. Female	0.78 (0.6–1)	0.06	240	na		
CEA	>3.5 mcg/L	1.45 (1–1.9)	0.02	206	1.4 (0.9–2.1)	0.1	206
CA19-9	>37 U/mL	1.2 (0.7–1.8)	0.3	220	na		
CRP	>0.6 mg/dL	1.7 (1.3–2.1)	<0.001	204	1.5 (1–2.1)	0.026	204
Leucocytes	>10 G/L	1.57 (1–2.1)	<0.006	236	1.4 (0.9–2.1)	1.1	236
Neutrophils	>7 G/L	1.7 (1.2–2.6)	0.003	171	1.3 (0.7–2)	0.8	171
Monocytes	>1 G/L	1.8 (1–3.2)	<0.039	171	1.2 (0.8–1.9)	0.2	171
NLR	>6 (ROC)	1.7 (1.2–2.4)	<0.001	171	1.7 (1.2–2.6)	0.003	171
LMR	>3.98 (ROC)	0.6 (0.4–0.8)	0.01	171	1.4 (0.9–2.1)	1.1	171
Plt/LR	>333 (ROC)	1.1 (0.7–1.7)	0.5	171	1.6 (1.–2.6)	0.2	171
Platelets	>400 G/L	1 (0.6–1.6)	0.9	236	na		

Abbreviations: ULN = upper limit of normal, cox regression analyses, Ci = confidence interval, na = not available, vs. = versus, NLR = neutrophils to lymphocytes ratio, LMR = lymphocytes to monocytes ratio, PLT/LR = platelets to lymphocyte ratio, G = giga, ROC = ROC Youden Analyses.

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
