# Peer review of "C-Reactive Protein and Neutrophil/Lymphocytes Ratio: Prognostic Indicator for Doubling Overall Survival Prediction in Pancreatic Cancer Patients"

_jcm, 2019, doi:10.3390/jcm8111791_

Round 1

Reviewer 1 Report

To the Authors

I have with great interest read this original article regarding CRP and NLR as a prognostic markers for pancreatic cancer. Since pancreatic cancer is a very severe disease and not cure is available for inoperable cases prognostic markers are crucial for administration of chemotherapy.

I think this is a good study that deserves publication, although some minor revisions would improve the manuscript.

Firstly in the text and abstract all abbreviations are not explained and although several are obvious, all needs to be defined the first time they are used. English is not my native language and I cannot give a full assessment of the language but there are a typos (Mai-May, therapie-theraphy) so a proofreading might be indicated Your found 367 pts and excluded 127 since they either were either cases of adjuvant or no therapy and had 240 for inclusion of which 56 were LAPC, was there no cases of neoadjuvant treatment or were they in the LAPC-group and is so did anyone got resected and what was then their NLR and crp? Especially since there was no difference in OS between metastatic disease and LAPC which differs from other studies one wonders which patients were in the LAPC-group and how they were defined

Regards

Author Response

Dear Editor,

We submit here with our revised manuscript entitled CRP and Neutrophil/ Lymphocytes ratio: A prognostic indicator for doubling OS prediction in Pancreatic Cancer Patients

We want to thank the reviewers for their helpful comments and further consideration of our manuscript. We addressed all issues raised in the manuscript and in our point-by-point-reply.

We thank the reviewer for the statement that this is an interesting and crucial study, given the recent developments with chemotherapy. We now address the reviewer’s remarks in our revision:

Reviewer #1:

Ad 1

In the text and abstract all abbreviations are not explained and defined the first time they are used.

We agree with the reviewer and double-checked all abbreviations in our revised manuscript.

Ad 2

English proofreading is indicated

We thank the reviewer for this valid point. We performed another round of in-depth proofreading and corrected all errors identified by this effort.

Ad 3

You found 367 pts and excluded 127 since they either were either cases of adjuvant or no therapy and had 240 for inclusion of which 56 were LAPC, was there no cases of neoadjuvant treatment or were they in the LAPC-group and is so did anyone got resected and what was then their NLR and crp? Especially since there was no difference in OS between metastatic disease and LAPC which differs from other studies one wonders which patients were in the LAPC-group and how they were defined

Our LPAC group was defined as “never- operable” at the time point of diagnosis by interdisciplinary tumor board according to NCCN guidelines, primary due to venous and or arterial involvement. Therefore, these patients received primary palliative chemotherapy.

We appreciate this comment about neoadjuvant treatment. This group of patients (n=6) did get operated on after receiving neoadjuvant chemotherapy and was included in the adjuvant cohort. However, these patients were not included in our analyses as mentioned in our manuscript.

We agree with the reviewer that one might expect a difference between LPAC and MPAC group regarding OS.

We hope that we sufficiently answered the questions, comments and suggestions.

If there are any questions, please do not hesitate to contact us.

With kind regards

Reviewer 2 Report

The authors addressed a very hotspot area of oncology research, which is the discovery of peripheral prognostic biomarkers that can be easily implemented in the majority of hospitals/clinics that handled cancer patients.

The patients’ cohort is good and allows an appropriate statistical analysis of the clinical and laboratorial results.

The manuscript is well written and the figures and tables are adequate.

Minor aspects:

In fig.1, or in a new figure, the authors should represent the OS for patients with each one of the prognostic factors and for those who had the 2 prognostic factors simultaneously. Moreover, OS should also be evaluated for the 2 prognostic (NLR and CRP), particularly in metastatic PC. In this new era of higher capability of cell characterization, for sure, it will be interesting, and since monocytosis was found, to quantify, at least, the 3 principal monocytes subpopulations (classical, intermediate and non-classical), as well as myeloid suppressor cells.

Author Response

Rebuttal letter: Response to reviewer

Dear Editor,

We submit here with our revised manuscript entitled CRP and Neutrophil/ Lymphocytes ratio: A prognostic indicator for doubling OS prediction in Pancreatic Cancer Patients

We want to thank the reviewers for their helpful comments and further consideration of our manuscript. We addressed all issues raised in the manuscript and in our point-by-point-reply.

We thank the reviewer for the statement that this is an interesting and crucial study, given the recent developments with chemotherapy. We now address the reviewer’s remarks in our revision:

Reviewer #2:

Ad 1.1

The authors should represent the OS for patients with each one of the prognostic factors (CRP and NLR) and for those who had the 2 prognostic factors simultaneously.

We appreciate this comment and are eager to provide further data regarding zero, one and two risk factors

Of the 145 patients with complete data available for all variables 45 patients had none of the above mentioned risk factors, 65 patients had one, and 35 had both risk factors. The median OS for all patients was 8.3 months.

Patients with zero risk factors had 11.3 months compared to 7.1 months with one and 3.8 months with two risk factors respectively. Survival differences among groups were statistically significant (HR=1.558; 95% CI= (1.26-1.95)); p<0.001).

We have added this figure to the manuscript.

Ad 1.2.

Moreover, OS should also be evaluated for the 2 prognostic (NLR and CRP), particularly in metastatic PC

Of the 123 MPAC patients with complete data available for all variables 35 patients had none of the above mentioned risk factors, 57 patients had one, and 31 had both risk factors. The median OS for all patients was 10.6 months.

Patients with zero risk factors had 9.8months compared to 5.7 months with one and 2.4 months with two risk factors respectively. Survival differences among groups were statistically significant (HR=1.5; 95% CI= (1.1-1.9)); p=0.003). We have added this information to the manuscript.

Ad 2

In this new era of higher capability of cell characterization, for sure, it will be interesting, and since monocytosis was found, to quantify, at least, the 3 principal monocytes subpopulations (classical, intermediate and non-classical), as well as myeloid suppressor cells.

This is indeed a very interesting question in order to better understand the contribution of monocytes to inflammation. However, this analysis was primary designed to focus on clinically available data and was done in a retrospective manner; no biobank of this cohort is available. Thus, we cannot address this question in our cohort

We hope that we sufficiently answered the questions, comments and suggestions.

If there are any questions, please do not hesitate to contact us.

With kind regards
